# Role of Dynamic Response in Inclined Transverse Crack Inspection for 3D-Printed Polymeric Beam with Metal Stiffener

**DOI:** 10.3390/ma16083095

**Published:** 2023-04-14

**Authors:** Arturo Francese, Muhammad Khan, Feiyang He

**Affiliations:** 1School of Aerospace, Transport and Manufacturing, Cranfield University, College Road, Cranfield MK43 0AL, UK; a.francese@cranfield.ac.uk; 2Centre for Life-Cycle Engineering and Management, School of Aerospace, Transport and Manufacturing, Cranfield University, College Road, Cranfield MK43 0AL, UK

**Keywords:** material extrusion, dynamic response, damage identification, stiffened cantilever beam, modal strain energy damage index

## Abstract

This paper aims to quantify the relationship between the dynamic response of 3D-printed polymeric beams with metal stiffeners and the severity of inclined transverse cracks under mechanical loading. Very few studies in the literature have focused on defects starting from bolt holes in light-weighted panels and considered the defect’s orientation in an analysis. The research outcomes can be applied to vibration-based structure health monitoring (SHM). In this study, an acrylonitrile butadiene styrene (ABS) beam was manufactured through material extrusion and bolted to an aluminium 2014-T615 stiffener as the specimen. It simulated a typical aircraft stiffened panel geometry. The specimen had seeded and propagated inclined transverse cracks of different depths (1/1.4 mm) and orientations (0°/30°/45°). Then, their dynamic response was investigated numerically and experimentally. The fundamental frequencies were measured with an experimental modal analysis. The numerical simulation provided the modal strain energy damage index (MSE-DI) to quantify and localise the defects. Experimental results showed that the 45° cracked specimen presented the lowest fundamental frequency with a decreased magnitude drop rate during crack propagation. However, the 0° cracked specimen generated a more significant frequency drop rate with an increased crack depth ratio. On the other hand, several peaks were presented at various locations where no defect was present in the MSE-DI plots. This suggests that the MSE-DI approach for assessing damage is unsuitable for detecting cracks beneath stiffening elements due to the restriction of the unique mode shape at the crack’s location.

## 1. Introduction

Increased demand for building complex and lightweight aerospace structures with lower development costs has given rise to the use of additive manufacturing (AM), such as 3D printing, as a robust manufacturing route. The aerospace industry has been one of the most promising sectors for using AM due to its reduced cost, greater design flexibility, and increased demand for greater product varieties. According to Wohlers’ annual report in 2017 [1], the aerospace industry accounted for 20% of the total AM demand. The shift from using AM parts as prototyping devices to more reliable and functional components became possible due to greater attention being paid to development in this area [2,3].

Two of the most common polymer-based additive manufacturing (AM) methods for aerospace applications are stereolithography (SLA) and fused deposition modelling (FDM). The former has been observed to produce sturdier parts than those created with FDM, and is, thus, better suited for casting aerospace parts. The latter has seen high-performance space applications, such as flame-retardant vents, housings, large pod doors, and complex electronics in the NASA Rover called Desert RATS [2]. FDM involves heating the material near the melting point to create a liquid-like layer and extruding it through a computer-controlled nozzle. Layer-by-layer bonding through thermal diffusion is possible due to the relatively high-temperature printing environment [3]. The production of complex shapes for aerospace structures has gained much interest due to its high strength-to-weight ratio, leading to significant mass reduction and high mechanical properties. Forcellese et al. [4] explored the environmental aspects and improved the strength of 3D-printed composite isogrid structures. Turner et al. [5] designed and manufactured 3D-printed sandwich structures with acrylonitrile butadiene styrene (ABS) material and Kevlar fabric as the face sheets. They observed an improvement in the impact strength, which was possible with the 3D printing of lattice structures and the selective placement of support trusses.

Complex aerospace 3D-printed structures necessitate a robust and reliable inspection method to monitor their behaviour throughout their service life under fatigue loading conditions. A visual inspection has always been the norm in ensuring the required level of safety and durability throughout the aircraft’s service life [6]. However, conducting a visual inspection is a labour-intensive activity with strict limitations. To solve this long-standing problem, structural health monitoring (SHM) aims to provide a real-time in situ assessment of the safety and reliability of an aircraft structure [7]. Techniques based on dynamic response have become a topic of significant interest due to their main advantages in localising and detecting damage, particularly for large components inaccessible for inspection [8,9]. Some methods have gained practical civil engineering applications in monitoring dams and bridges [10,11,12,13]. With advanced noncontact measurement techniques, such as 3D laser vibrometry, the experimental modal analysis could also be conducted for lightweight structures [14,15]. However, there are fewer practical aerospace applications, mainly due to the intrinsic nature of aerospace restrictions and safety regulations, leading the aerospace industry to have yet to embrace this method. Other factors, such as the number and placement of sensors, weight concerns, and the reliability of the sensor systems, should be the main focus for future developments, where 3D-printed smart and multifunctional materials have the capabilities necessary to carry out this role as an integrated health monitoring system [2].

The modal-based damage assessment relies on detecting changes in the dynamic response of a structure influenced by the mechanical properties of the system, such as stiffness, mass distribution, and damping, as well as the presence of damage in the structure [9,16]. These changes can be used to assess damage by comparing the current state of the structure to its initial or previous condition. Rytter [17] introduced four damage assessment levels, each requiring a specific approach to dynamic response models: level I for damage detection, level II for damage localisation, level III for damage severity quantification, and level IV for the remaining life prediction. Two main models are used in the modal-based damage assessment: the direct modal-based model, which uses modal parameters directly, and the extended modal-based model, which uses derived modal parameters [18]. This paper focuses mainly on levels I and II, as the experimental setup allows for two measured parameters during tests.

A beam structure was selected in the research to avoid introducing complexity into the model due to geometry. Beams are widely used in engineering, so this choice should not have negatively impacted the general applicability of the model. In the presence of a crack, a beam structure has been shown to produce a different dynamic response compared to a baseline, which can be used to extract natural frequency shifts through a direct modal-based model for damage detection and a localisation assessment [9,19,20,21,22,23]. However, this approach tends to be less sensitive to obtaining damage localisation (level II) than an extended modal-based approach, which mainly uses modal flexibility, modal curvature, and its derived modal strain energy [18]. Govindasamy et al. [24] used two extended modal-based damage indicators, namely, normalised modal shape curvature and modal strain energy, to obtain level I damage detection, with better performance found for the modal shape curvature method. To obtain both a level I and II damage assessment, Kahya et al. [25] developed a FEM model of open transverse edge cracks in laminated composite beams by employing changes in the mode shape curvature and the modal flexibility based on experimental and numerical modal data. The FEM model results were in good agreement with the experimental data, perhaps because the beam model is simple. However, the main limitation that needs to be addressed is the performance of the tools when applied to more complex structures, different geometries, and boundary conditions. For example, limited research has been conducted on using vibration-based structural health monitoring (SHM) for bolted connections with defects in aircraft structures [26,27]. Although several studies have investigated the behaviour of damage initiation and propagation in the vicinity of bolts [27,28], as well as the dynamic response of loosened bolts in pipelines and aerospace structures [29,30], the field of damage assessments has yet to be extensively explored with respect to cracked bolt holes.

One previous piece of research explored vibration-based damage detection tools for composite skin-stiffener structures [18,26]. Specifically, extended modal-based models employing strain energy-based indicators successfully achieved a level I and II damage assessment, with the addition of a level III damage assessment through the combination of direct and extended modal-based algorithms. The study effectively demonstrated the capacity for accurately evaluating levels I and II on a complex composite structure comprising composite skin and multiple stringers. However, it failed to quantitatively estimate the damage severity (level III) due to the damage index’s mathematical rather than physical nature [18]. To provide a valuable structural health monitoring tool for aircraft composite structures, it is imperative to enhance its capability from damage detection and localisation (levels I and II) to damage severity (level III) and a life prediction assessment (level IV).

Following the findings in [16,24], it is acknowledged that an ABS cantilever beam with a metallic stiffener serves as a physical representative model to quantify the relationship between changes in the dynamic response of an intact and defective structure under the same fatigue loading conditions. This arrangement, which behaves like a beam model, may be readily expanded to a plate, thus, increasing the model’s flexibility and generality.

However, all the studies mentioned above did not consider the initial crack orientations, which is hardly the perfect transverse in actual cases. Moreover, very few studies in the scientific literature focused on a defect starting from a bolt hole in a light-weight panel. Therefore, this study emphasises the investigation of the specific behaviour of the ABS skin sample, while a metallic stiffener is generally used to add local stiffness to a structure applied in an aerospace structure. The ABS skin was seeded with different orientation cracks from a bolt hole towards the closest sample edge to simulate a typical defect. The damage was seeded having different initial depths and orientations concerning the direction of the applied mechanical loading.

## 2. Materials and Methods

### 2.1. Specimen Schematic and Specification

This study focused on the experimental evaluation of diverse seeded cracks on a cantilever beam manufactured from ABS material, both with and without a metallic stiffener. The nomenclature of each damage scenario is demonstrated in Figure 1, and each specimen was assigned a corresponding code. An intact specimen was employed to establish a baseline for the dynamic response parameters. The seeded crack was characterised by an initial depth and orientation with respect to the direction of the applied loading, which was exerted at the free end of the beam to maximise the applied tension at the crack tip and minimise testing time. The seeded crack began close to the edge of the first bolt hole. This location was chosen due to the stress concentration resulting from the applied bending moment at the beam’s free end, considering that bolt holes are typical locations for crack initiation.

Natural frequency changes and local strain measurements were collected at ambient temperature to establish the relationship between various crack scenarios and their dynamic response. Before conducting each experiment, the specimens underwent meticulous inspection for any significant surface defects. It was essential to ensure that the sample performed optimally during the vibration test to avoid any potential influence of external parameters. Moreover, this study assumed that no interaction existed between the crack propagating from the seeded crack and a typical stress concentration factor at a bolt hole.

### 2.2. Specimen Preparation

The specimen comprised a 3D-printed ABS skin and an aluminium alloy (AA) 2014-T615 stiffener. The effective length of the specimen was 200 mm with a 3.1 mm skin thickness. The main dimensions of the hybrid 3D-printed beam and the metallic stiffener are shown in Figure 2. The mass of the ABS skin was approximately 35 g. The mass of the stiffener was 45.2 g. The skin and stiffener were joined with a single line of 11 fasteners along the specimen’s slender section. The total mass of the assembled specimen with bolts and nuts was approximately 85 g. The sample was clamped at its fixed end with four bolts and sandwiched between two metallic plates, as shown in Figure 3. To simulate the presence of a defect, a lateral crack was seeded on the ABS skin on the side of the first bolt hole at a 10 mm distance from the stress concentration fillet, also shown in Figure 2. The crack location was intentionally chosen to be close to the fixed end to prevent the high-stress concentration moving away from the crack location. Two increasing crack depths were selected for the defective samples, 1 mm and 1.4 mm with 0°, 30°, and 45° angle orientations for each crack depth, as indicated at the bottom of Figure 2. The crack width was also fixed at 0.2 mm to ensure manufacturability. The crack was manufactured using a sharp razor tool to its specified dimension. A DinoCapture camera was used to inspect whether the actual size of the crack was manufactured accordingly. Moreover, it is worth emphasising that only the ABS material had a seeded crack.

The ABS skin samples were 3D printed with a Raised3D Pro 2 (Raise 3D Technologies Inc., Irvine, CA, USA) industrial-grade printer. The CAD model of the specimen was created in CATIA V5 (Dassault Systèmes, Vélizy-Villacoublay, France) and saved in STL format to enable importing the model into ideaMaker 4.1.1. software used with Raised3D Pro 2 for the printing preparation. Red ABS skin was used to ensure easier crack detectability, especially during its crack propagation.

The 3D printing parameters played an important role in the mechanical strength of the sample [31]. Based on the previous research by Baqasah et al. [3], it was suggested that a ±45° build orientation be used to print the sample. This was to consider the optimum trade-off of the weaker strength in the lateral direction and greater strength in the longitudinal direction to the test direction [32,33]. Table 1 summarises the printing parameters used for the ABS skin.

The stiffener was machined in aluminium alloy AA2014-T615 to the required dimensions. The stiffener was not investigated on its dynamic response change, as it was merely introduced to the sample to increase its geometrical complexity and stiffness.

### 2.3. Experimental Setup

The experimental setup is shown in Figure 3 and Figure 4. The fixed end of the sample was clamped between two metallic plates composed of stainless-steel plates and screwed in using four M4 bolts with washers and nuts to avoid high-stress concentration effects on the bolts. The support was attached to the shaker via two bolts, and it had a flat surface on which the sample could be mounted. The design of the stiffener was tapered at its end to provide further tolerance of the sample’s movement during its excitation. The sample was clamped on top of the shaker’s shaft. The bolts and nuts were tightened with a handheld torque driver. After clamping, the sample was checked on fixation by shaking it momentarily.

The vibration system consisted of a signal generator (AFG-2105), a 300 W power amplifier from Data Physics and a shaker or modal exciter originating from Data Physics (Data Physics, Santa Clara, CA, USA). A voltage signal of 5 V was used on the signal generator throughout the test schemes. The shaker was fixed to the ground to ensure stable output data only from the input parameters. An accelerometer, Brüel & Kjær 4535-B, with a frequency range from 0.3 Hz to 10 kHz, was mounted to the tip of the sample using a fixed bolt. Its mass was 6.0 g, which was assumed to be negligible concerning the specimen’s total mass (85 g). The accelerometer was connected to a data acquisition card NI-9234 through an NI cDAQ-9174 chassis from National Instrument. The test data were acquired and sent to a laptop for acquisition and postprocessing using NI SignalExpress© 2015 version from National Instruments (National Instruments, Austin, TX, USA) and MATLAB© (MathWorks, Natick, MA, USA) for the postprocessing.

RS Pro wire lead strain gauges were also used to measure the strain under a specific static load to generate a correlation between the experimental results and the model output. They were RS Pro wire lead strain gauges and were placed close to the seeded crack on the sample, as shown in Figure 4.

### 2.4. Experimental Procedures

The experiment started with the performance of impact and sweep tests to find the first natural frequency. An impact test was performed three times for each sample to obtain an average natural frequency value. A sweep test was also performed for each sample between a close range to the average value obtained through the tap test for further validation. Then, the shaker was run at the sample’s natural frequency to apply the maximum loading and minimise testing time. The sample was excited along the z-axis, and the transverse vibration mode was analysed, as shown in Figure 3. The output data consisted of measurements from the accelerometer close to the beam’s free end, representing the dynamic response from the specimen. The data were continuously collected using SignalExpress and visualised on screen in real-time during the experiment. After observing a drop in its dynamic response, i.e., in amplitude, a change in the specimen’s stiffness was recognised [19,20]. At this point, the test was stopped, and the crack length was captured with the DinoCapture camera. Subsequently, the test was restarted by searching for the new natural frequency. This process was repeated until the crack propagated entirely or showed no further propagation, which was assumed by no new drop in amplitude. The aim was to formulate a function of the dynamic modal response with respect to the change in crack length during the propagation phase.

### 2.5. Development of the FE Model

A 3D mesh of the sample with tetrahedron elements was built in ANSYS Workbench© 2020 r2 (ANSYS, Canonsburg, PA, USA). The influence of the mesh size was studied using a convergence analysis, and a mesh size was defined as an element size of 1 mm. A mesh option in Workbench, Body of Influence, allowed for the refinement of the mesh around a particular area. In the present case, it was used to refine the mesh around the crack tip while leaving an acceptable mesh in the rest of the model. The model was clamped at the bolt locations and a vertical acceleration to the body was applied to activate the model analysis. The first mode shape (amplitude) was investigated in the numerical model, and the results output was extracted to obtain the appropriate strain data to feed into the damage index formulation.

### 2.6. Modal Strain Energy Damage Index (MSE-DI) Method

Loendersloot et al. [18] proposed an extended modal-based model with a strain energy-based indicator to study vibration-based damage detection tools on composite skin-stiffener plates.

Firstly, a one-dimensional formulation of the strain energy was given in Equation (1):(1)$$U=\frac{1}{2}\int_{0}^{l}\left[EI_{y}\cdot {\left(\frac{\partial^{2}u_{z}\left(x\right)}{\partial x^{2}}\right)}^{2}\right]dx$$
where U is the strain energy, l is the beam’s length, $EI_{y}$ is the bending rigidity, and $u_{z}\left(x\right)$ is the displacement in the z-direction with respect to the x-coordinate.

The modal strain energy damage index (MSE-DI) has the powerful capability of localising damage without knowing which modes are the most sensitive. The drawback is that the methods not influenced by the damage can dampen the value of the damage index β [18]. There are many damage index variants using the modal strain approach, but only Equation (2) was considered for the present study [18,26,34]:(2)βj=∑n=1Nfreq[γ˜j(n)γ˜(n)]∑n=1Nfreq[γj(n)γ(n)]
(3)$$Z_{j}=\frac{\beta_{j}-\mu }{\sigma }$$

In Equation (2), $\beta_{j}$ is the damage index per element utilising the fractional value of the damaged structure (i.e., the variables indicated with the hat) and the baseline structure. The variables $\gamma_{j}^{\left(n\right)}$ and $\gamma^{\left(n\right)}$ represent the integral of the second term of Equation (1) over the element j and the entire structure length for each mode shape, respectively. Again, the ones with the hat indicated the fractional value of the damaged structure. A normalised value of the damage index shown in Equation (3) successfully used to demonstrate the capabilities of an accurate assessment of levels I and II on a more geometrically complex structure consisting of composite skin and multiple stringers [18]. In this study, the efficiency of the damage detection and localisation of the MSE-DI approach was investigated for a hybrid structure under fatigue loading.

## 3. Results

### 3.1. Modal Strain Energy Damage Index (MSE-DI)

The MSE-DI algorithm was implemented in MATLAB to generate a damage index of defective structures using the strain intensity data obtained from the FEM harmonic analysis of the first mode only. The damage index produced from the algorithm was normalised over the length of the structure to attenuate the noise and maximise the peak generation at the damage location.

#### 3.1.1. Hybrid-Stiffened Model

Three different damage scenarios were investigated using the hybrid-stiffened model, with the resulting modal analysis of the first mode shown in Table 2. The plotted normalised damage index is shown in Figure 5 and Figure 6.

#### 3.1.2. ABS Beam without Stiffener

The implementation of the damage index on the simplified ABS sample for 0° and 30° is shown in Figure 7(a.1,b.1), respectively. Figure 8(a.1) show the 3D plot of the normalised damage index of a 45° crack orientation. In addition, 2D plots of the sample’s length vs. normalised damage index are shown in Figure 7(a.2,b.2) and Figure 8(a.2) for a detailed damage detection indication. Three different damage scenarios were investigated using the hybrid-stiffened model, with the resulting modal analysis of the first mode shown in Table 3.

Based on these plots, it could be inferred that peaks appeared at particular locations with a significantly higher normalised damage index than the rest. Several false positives or peaks appearing at locations with no defects could also be seen.

### 3.2. Crack Propagation Experimental Data

The crack depth (CD) ratio vs. natural frequency drop were recorded for all damage scenarios until the sample failed with no further crack growth.

The CD ratio was obtained using Equation (4):(4)$$CD\;Ratio=\frac{Crack\;Depth}{Baseline\;Crack\;Depth}$$

In addition to the frequency drop result, as shown in Figure 9, Table 4 shows all fundamental frequencies measured in the experiments.

The crack depths were measured with a DinoScope digital microscope and recorded for each frequency drop. Figure 10 shows the crack sizes captured at the samples’ failure for all damage scenarios (i.e., 0°, 30°, and 45° crack orientations).

## 4. Discussion

### 4.1. Influence of Crack Orientation on the First Natural Frequency

Different dynamic responses were observed based on the experimental setup for each damage scenario. Table 4 shows all fundamental frequencies measured during the experiments. The differences in the magnitude of the first natural frequency with the crack orientation can be observed in Figure 9.

The 45° cracked samples presented the lowest first natural frequency compared to the 0° and 30° cracked samples. This may be explained by the amount of volumetric defects present in the cracked samples that were higher than 0° and 30°, resulting in more degradation to the sample’s stiffness [35]. In addition, ductile materials, such as ABS, experience highest shear stress at a 45° orientation, with the possible contribution of this effect shown by the magnitude of the first natural frequency.

On the other hand, no correlation between the 0° and 30° crack orientations and the resulting first natural frequency could be found, as shown in Figure 11. It is important to mention that additional crack depth data would be needed to obtain a more accurate and complete picture of the influence of the crack orientation on the resulting first natural frequency.

### 4.2. Modal Strain Energy Damage Index (MSE-DI)

This study implemented modal strain energy to detect and localise damage in assessing damage through experiments and the data postprocessing of a hybrid-stiffened panel and simplified skin structure.

#### 4.2.1. Hybrid-Stiffened Model

Based on the normalised damage index results of the stiffened panel shown in Figure 5 and Figure 6, it could be concluded that no accurate damage detection or localisation were present. The defect was generally indicated by peaks at the location of the damage. However, the peaks were present at various locations without defects in the plots. The differences in the strain intensity distribution on the stiffened panel and the skin of the sample may explain this. The stiffener effectively restricted the mode shape of the crack location indicated by the almost constant strain values. Consequently, this resulted in a noneffective damage index, suggesting that the MSE-DI approach may not be suitable for detecting cracks beneath stiffening elements. Furthermore, the noise at various sample locations was probably amplified into false positives.

#### 4.2.2. Simplified ABS Skin Model

Typically, the peaks present at the location of damage could be observed in Figure 7 and Figure 8, indicating the achievement of more effective damage detection and localisation compared to the hybrid-stiffened panel. However, some false positives still occurred at several locations, albeit less frequently than the result of the hybrid-stiffened panel. This result was expected due to the unique mode shape of the sample indicated by more pronounced differences in the strain intensity at the crack tip. This strain intensity distribution also attenuated some of the noise observed in the stiffened panel sample due to the normalisation procedure performed. In order to provide more accurate damage localisation and reduce the number of false positives, higher bending modes could be included within the damage assessment algorithm [18].

### 4.3. Crack Propagation Rate and Path

Based on the experimental propagation results, some correlations were observed between the magnitude of the frequency drop and crack orientation. It was observed that the 0° crack orientation samples generated a greater magnitude of frequency drops after an increase in the CD ratio. This could be explained by the higher stress intensity factor at the crack’s tip initiating a more rapid crack growth due to the bending mode of the first natural frequency. On the other hand, the 30° and 45° cracked samples showed a decrease in the frequency drop magnitude after an increase in the CD ratio. This may have possibly occurred due to the smaller magnitude and coverage of the stress intensity factor (SIF) inducing a less severe frequency drop after each successive drop.

Notably, the fracture of the 45° cracked samples initiated at the right side of the sample and grew towards the seeded crack, as schematically represented in Figure 12. This was possibly caused by a higher stress concentration at the initiation location of the sample, compared to the seeded crack location. This also gave some evidence that the bending stress within this test scheme played a more significant role than the shear stress in generating the SIF and, consequently, initiating a defect at the right side of the sample. Furthermore, the propagation path was shown to be influenced by the crack orientation because the propagation path was almost linear for samples with a 0° crack orientation, and a nonlinear behaviour was observed at 30° and 45° cracked samples, as seen in Figure 10.

## 5. Conclusions

Quantifying the influence of defects on changes in the dynamic response of complex structures brought clear insight into the practicality and robustness of structural health monitoring (SHM) for aerospace applications. The present work particularly was one of the first attempts to investigate oblique defects starting from the bolt holes.

The experimental tests were conducted using a cantilever beam, 3D printed using ABS material, with a metallic stiffener joined on top of the beam. An empirical investigation was carried out by correlating different damage scenarios to the local stiffness, propagation rate, and propagation path; a damage index was then proposed for damage detection and localisation. The main findings of the present study were:The specimen with a 45° crack presented the lowest first natural frequency compared to the ones with 0° and 30° cracks. No correlation was proposed for the latter two on the resulting first natural frequency.The MSE-DI approach for the damage assessment of stiffened cantilever beams was not successful in giving an accurate solution for damage detection and localisation due to several peaks being present at various locations where no defect was present. This suggests that the MSE-DI approach for conducting damage assessments is unsuitable for detecting cracks beneath stiffening elements due to the restriction of the unique mode shape at the crack location.The MSE-DI approach for the damage assessment of ABS cantilever beams with no stiffener performed more effectively in detecting and localising damage. Peaks were present at the damaged locations. However, some false positives still occurred, albeit less frequently. This was possibly due to using the first bending mode only. Perhaps adding higher modes would increase the accuracy of the model.The specimen with a 0° defect generated a higher frequency drop rate with a higher CD ratio. On the other hand, the 30° and 45° ones showed a decrease in the frequency drop rate, with an increase in the CD ratio.The propagation path seemed to be strongly influenced by the crack orientation. The 0° cracked samples showed a linear propagation path, while the 30° and 45° cracked samples clearly showed a nonlinear behaviour, very close to a parabolic shape.

## Figures and Tables

**Figure 1 materials-16-03095-f001:**
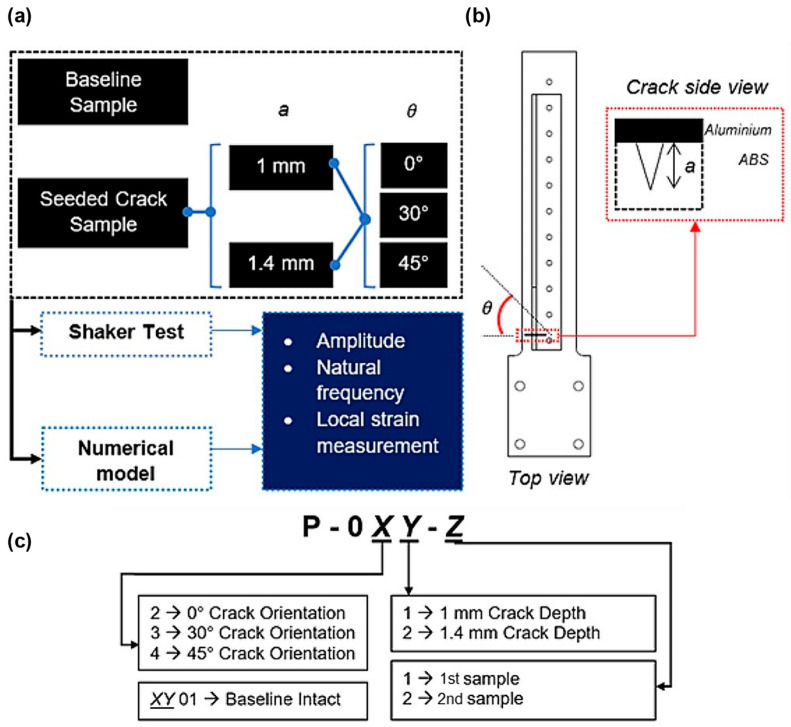
(**a**)Damage scenario, (**b**) crack location on the sample, and (**c**) sample codification.

**Figure 2 materials-16-03095-f002:**
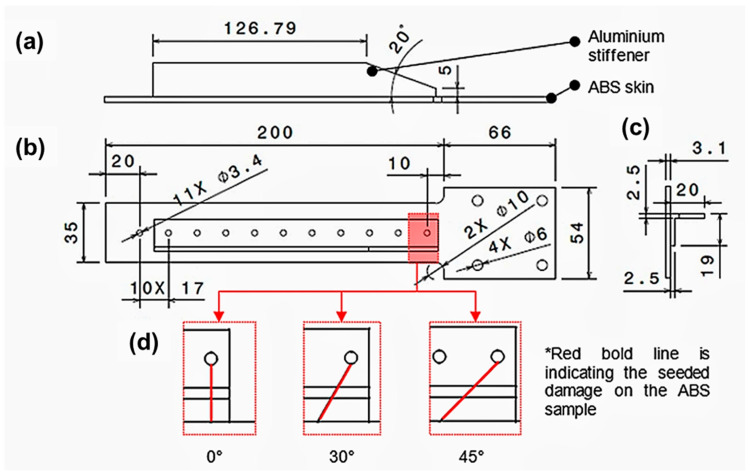
Specimen dimensions: (**a**) side, (**b**) top, and (**c**) cross-sectional view; (**d**) damage orientation from top view. Red line from the bolt represents the initial-seeded crack orientation.

**Figure 3 materials-16-03095-f003:**
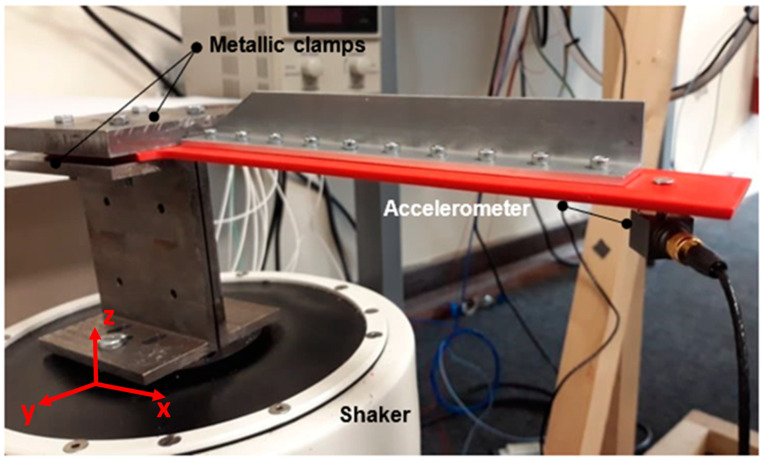
Clamping configuration of the sample.

**Figure 4 materials-16-03095-f004:**
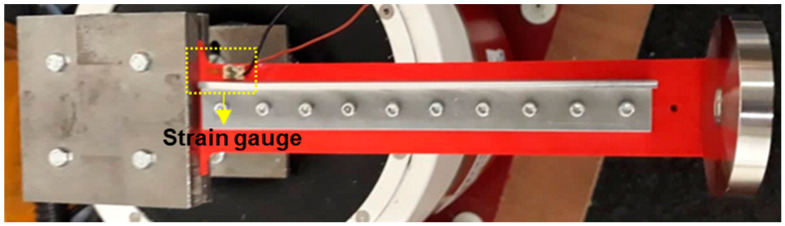
Strain gauge instrument and placement on the sample.

**Figure 5 materials-16-03095-f005:**
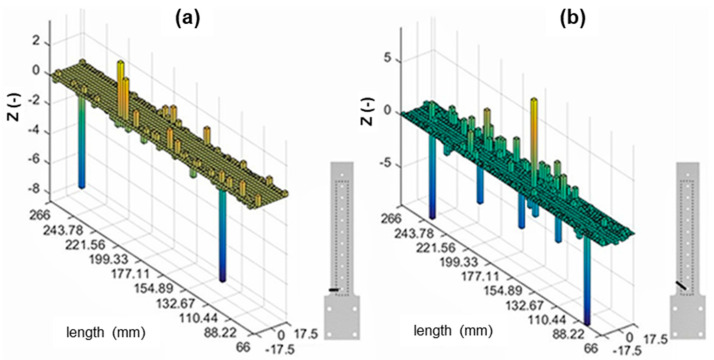
MSE-DI implementation (bottom layer) to hybrid-stiffened sample: (**a**) 0° cracked and (**b**) 30° cracked samples.

**Figure 6 materials-16-03095-f006:**
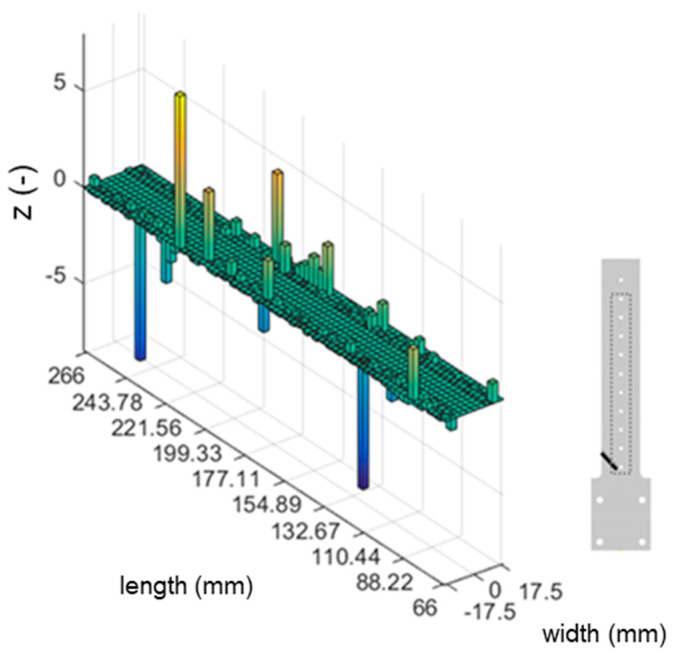
MSE-DI implementation (bottom layer) to hybrid-stiffened sample: 45° cracked sample.

**Figure 7 materials-16-03095-f007:**
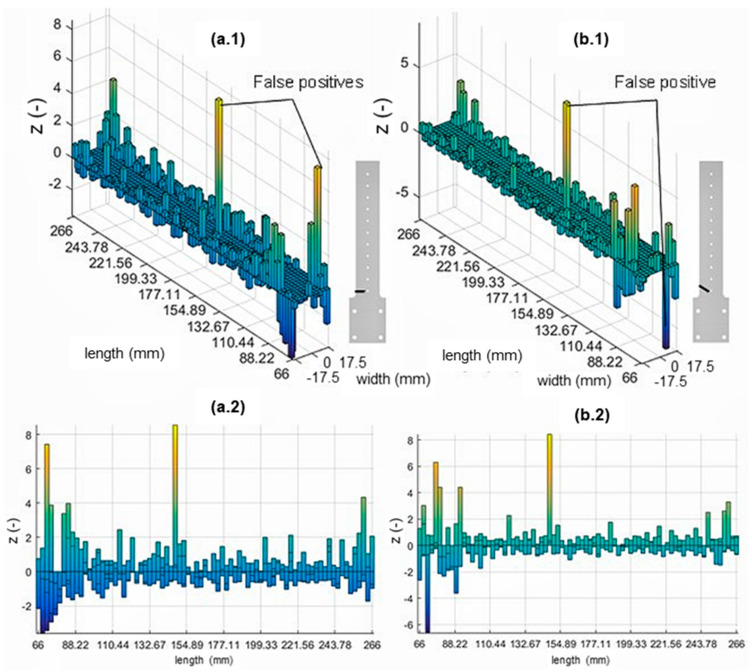
MSE-DI implementation to simplified ABS sample (skin only): (**a.1**,**a.2**) 0° cracked and (**b.1**,**b.2**) 30° cracked samples.

**Figure 8 materials-16-03095-f008:**
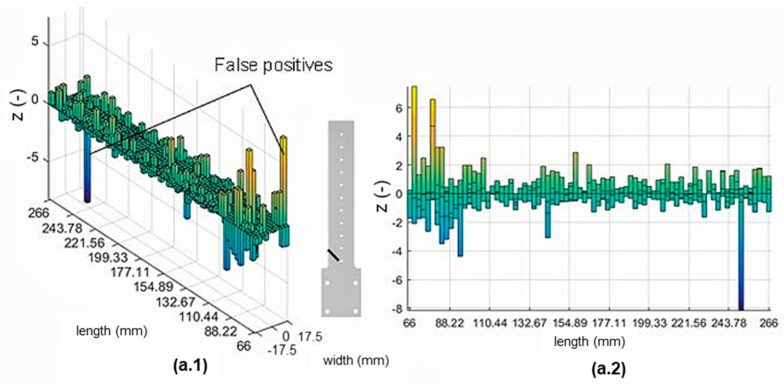
MSE-DI implementation to simplified ABS sample (skin only): (**a.1**,**a.2**) 45° cracked sample.

**Figure 9 materials-16-03095-f009:**
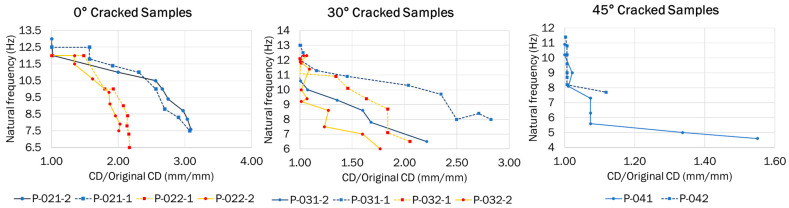
Crack depth ratio (CD/Original CD) vs. natural frequency for 0°, 30°, and 45° crack orientation.

**Figure 10 materials-16-03095-f010:**
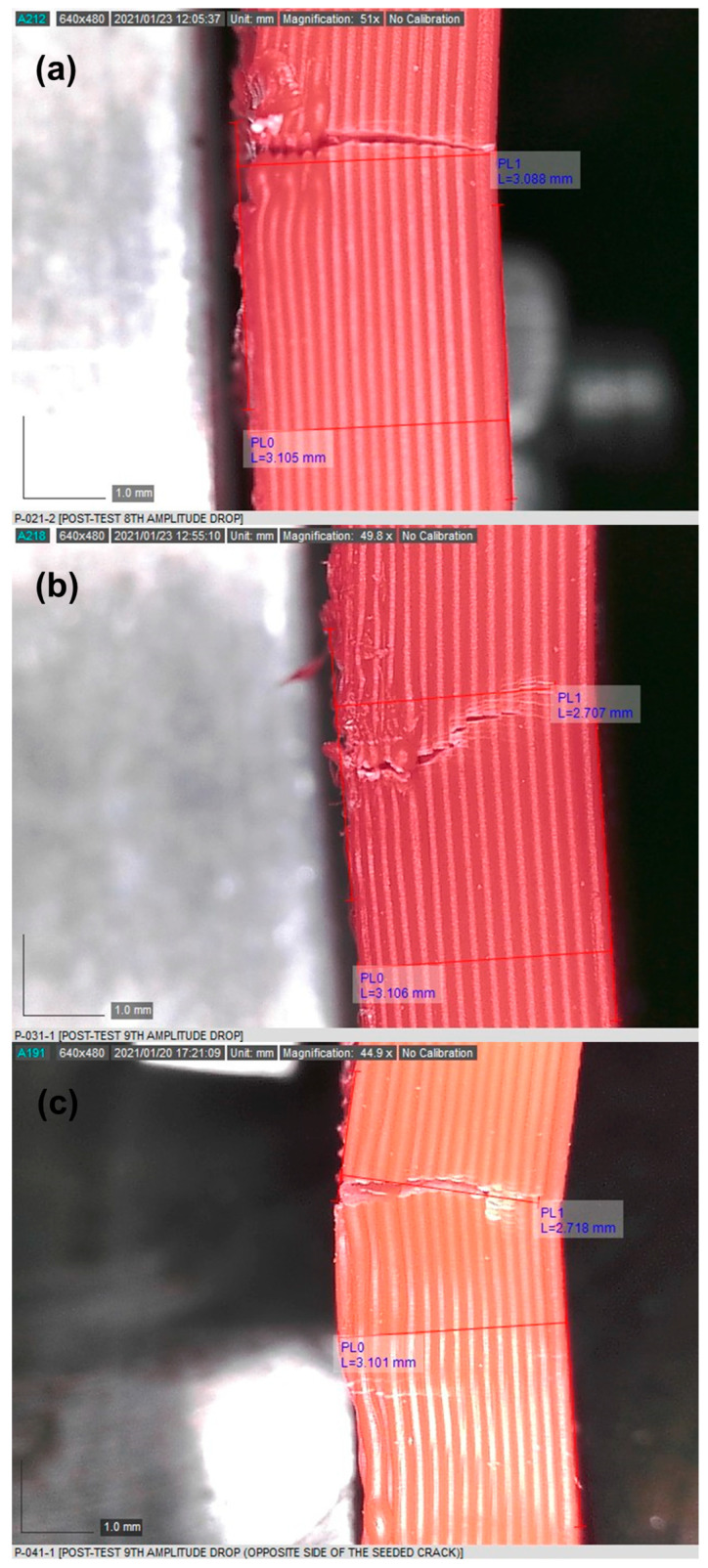
Crack depth of samples at failure: (**a**) 0° cracked, (**b**) 30° cracked, and (**c**) 45° cracked.

**Figure 11 materials-16-03095-f011:**
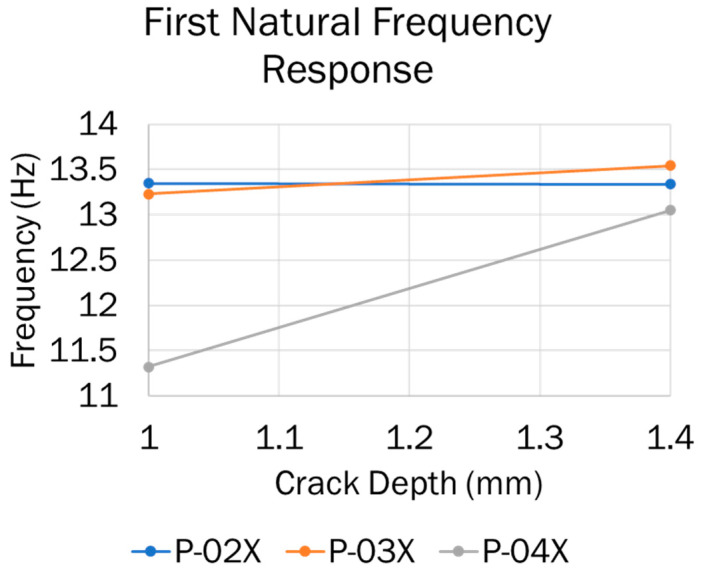
The first natural frequency of samples with various damage scenarios.

**Figure 12 materials-16-03095-f012:**
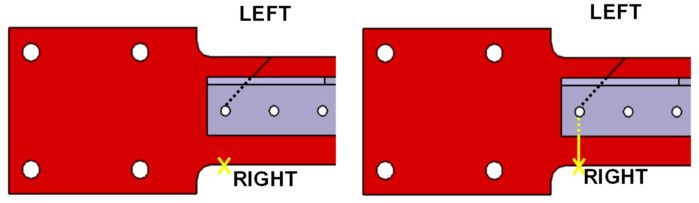
Crack growing from the opposite side of the 45° cracked sample. The yellow x indicates the start point of the crack propagation. The yellow line represents the crack path.

**Table 1 materials-16-03095-t001:** Printing parameters of the ABS skin.

Parameters	Value
Nozzle size	0.4 mm
Layer height	0.2 mm
Infill density	100%
Print orientation	±45°
Print speed	50 mm/s
Extruder temperature	250 °C
Bed temperature	100 °C
Wall thickness	2 mm

**Table 2 materials-16-03095-t002:** First mode natural frequency of MSE-DI hybrid-stiffened models.

Sample ID	Damage Scenario	Mode 1 Natural Frequency (Hz)
P-021	1 mm crack depth; 0° orientation	14.11
P-031	1 mm crack depth; 30° orientation	14.14
P-041	1 mm crack depth; 45° orientation	14.17

**Table 3 materials-16-03095-t003:** The first mode natural frequency of MSE-DI simplified ABS skin models.

Sample ID	Damage Scenario	Mode 1 Natural Frequency (Hz)
P-021	1 mm crack depth; 0° orientation	9.78
P-031	1 mm crack depth; 30° orientation	9.8
P-041	1 mm crack depth; 45° orientation	9.81

**Table 4 materials-16-03095-t004:** Summary of the first natural frequency of samples with different damage scenarios.

Experimental ID	Orientation (Degree)	Crack Depth (mm)	Experimental Mode 1 (Hz) (Average)
P-001	Intact	Intact	14.68
P-021	0	1.0	13.35
P-022	0	1.4	13.35
P-031	30	1.0	13.23
P-032	30	1.4	13.55
P-041	45	1.0	11.32
P-042	45	1.4	13.05

## Data Availability

Data available on request from the authors.

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
