# Peer review of "Role of Dynamic Response in Inclined Transverse Crack Inspection for 3D-Printed Polymeric Beam with Metal Stiffener"

_materials, 2023, doi:10.3390/ma16083095_

Round 1

Reviewer 1 Report

Dear authors,

I would like to congratulate you on a potentially very interesting paper with a comprehensive approach to developing methodologies to evaluate material defects.

A)    General remarks

1.      The article is clearly written and easy to follow.

2.      The authors give relevant references which are linked to their study

3.       The abstract is well written introducing the basic overview of the paper. It is also written in a way that even a person not familiar with the topic can understand what the authors are proposing in their research. Would suggest to emphasis strongly the novelty of the study. This is a key problem of the article.

4.      The introduction requires some improvements. The purpose of the introduction is to present the problem of the article and clearly present the overview of the state of the art in case of the topic and presented later and methods. Some problems noticed:

a.       The introduction is rushed. The background of novel testing techniques of especially light materials (eg. Moda analysis) is not presented. Authors use a classical approach with the modal techniques but there are much better alternatives which must be mentioned in the state of art analysis. Techniques ideal for the such task include e.g 3D laser vibrometry e.g https://doi.org/10.18429/JACoW-IPAC2018-WEPMF079

b.      Internal damage detection and the use of modal analysis that is also a great tool for quality assurance and control. Some examples of possible applications also for lightweight structures DOI: 10.3390/s23031263

c.       Line 42 “lab condition” – please do not use a common language (laboratory conditions). Please check the whole article for other such problems. Please use precise language.

d.      In the last paragraph were aim and scope are presented, however, the novelty is not presented strongly enough (in the scope of state-of-art evaluation). This part of the introduction must be presented in a clear and definite way. In its current state, it requires changes.

5.      Material and methods:

a.       What is the sample mass

b.      What is the accelerometer mass

c.       Assuming that the sample is a lightweight structure adding the mass of the accelerometer will change the dynamic condition of the test. Please include an explanation to the reviewer and place it in the text

d.      No specific data on some of the test equipment.

6.      The results discussion is adequate.

7.      Conclusions are acceptable. However, an emphasis on novelty would be profitable. The authors do not mention how the research will advance the field, where can it be used and what are the next steps. This is a serious problem of the article.  

B)      Items remarks

Fig.1 is out of focus. Maybe do to compression to the pdf. Please check before submitting the revised version.

Fig.2 please enlarge so everything is visible and the text is readable.

Fig.4 would suggest adding a picture (b) with an enlarged strain gauge and its placement.

Fig.5 is out of focus (especially the axis with the values). Maybe due to compression to the pdf. Please check before submitting the revised version.

Fig.7 is out of focus (especially the axis with the values in fig a2 and b2). Maybe due to compression to the pdf. Please check before submitting the revised version. Similar problem Fig.8

C)      Conclusions

The article is clear and interesting with no significant errors found in the research, however, the question of adding the mass of the accelerometer must be answered. Both methodology and results acquisition is correct. Some additional state-of-the-art analysis of modern testing techniques has to be incorporated in the introduction. The article has potential after eventual improvements. There are numerous small things to improve, however, at the current stage, the reviewer asks for minor changes in those areas and will be happy to accept the paper after those corrections.

Reviewer 2 Report

Authors of this manuscript study special features for crack detection in cantilever beam manufactured by 3D printing. The topic is relevant for the SHM community. Despite the fact that the investigation looks as unfinished, the presented in manuscript results can be published like intermediate but noteworthy.  I have following remarks to the manuscript, which need to be addressed before its publication.

1.       The labels for references are not correctly displayed.

2.       What are gammas with wave in Eq. (2)? You did not explain these notations.

3.       You applied the FEM harmonic analysis in your investigation. Please describe the soft that was utilized. What kind of mesh and its size are used? Also it would be usefull for validation and confidence in results, if you evaluate the numerical error of the computed by FEM data.     

Round 2

Reviewer 3 Report

Thank you for addressing my comments.

I wonder if the authors try to say “activate the modal analysis”, not “activate the model analysis” on line 256.